# Fertilization Type Differentially Affects Barley Grain Yield and Nutrient Content, Soil and Microbial Properties

**DOI:** 10.3390/microorganisms12071447

**Published:** 2024-07-17

**Authors:** Stefan Shilev, Anyo Mitkov, Vanya Popova, Ivelina Neykova, Nikolay Minev, Wieslaw Szulc, Yordan Yordanov, Mariyan Yanev

**Affiliations:** 1Department of Microbiology and Environmental Biotechnologies, Agricultural University—Plovdiv, 4000 Plovdiv, Bulgaria; vv_girl@mail.bg (V.P.); ivababrikova@abv.bg (I.N.); 2Department of General Agriculture and Herbology, Agricultural University—Plovdiv, 4000 Plovdiv, Bulgaria; anyomitkov@abv.bg (A.M.); mar1anski@abv.bg (M.Y.); 3Department of Agrochemistry and Soil Science, Agricultural University—Plovdiv, 4000 Plovdiv, Bulgaria; nikiminev@abv.bg; 4Department of Soil Environmental Sciences, Warsaw University of Life Sciences—SGGW, 02-787 Warsaw, Poland; wieslaw_szulc@sggw.edu.pl; 5Department of Crop Science, Agricultural University–Plovdiv, 12 Mendeleev Str., 4000 Plovdiv, Bulgaria; j.jordanov72@gmail.com

**Keywords:** barley, vermicompost, biochar, mineral fertilizer, organic fertilizer, soil microbial activity, grain yield, soil nutrients

## Abstract

The use of artificial fertilizers follows the intensification of agricultural production as a consequence of population growth, which leads to soil depletion, loss of organic matter, and pollution of the environment and production. This can be overcome by increasing the use of organic fertilizers in agriculture. In the present study, we investigated the effect of using vermicompost, biochar, mineral fertilizer, a combination of vermicompost and mineral fertilizer, and an untreated control on alluvial-meadow soil on the development of fodder winter barley *Hordeum vulgare* L., *Zemela cultivar*. We used a randomized complete block design of four replications per treatment. Barley grain yield, number of plants, and soil and microbiological parameters were studied. We found statistically proven highest grain yield and grain protein values when applying vermicompost alone, followed by the combined treatment and mineral fertilizer. The total organic carbon was increased by 70.2% in the case of vermicompost and by 44% in the case of combined treatment, both compared to the control. Thus, soil microbiome activity and enzyme activities were higher in vermicompost treatment, where the activity of β-glucosidase was 29.4% higher in respect to the control, 37.5% to the mineral fertilizer, and 24.5% to the combined treatments. In conclusion, our study found the best overall performance of vermicompost compared to the rest of the soil amendments.

## 1. Introduction

The increase in population has caused a rise in pesticide and artificial fertilizer applications to produce more food and feed that satisfies the need. A serious drawback is that inputting more chemicals in agroecosystems may result in the pollution of soils and waters [1,2]. At the same time, the circular approach is fundamental in the EU’s strategy to protect the environment, decrease greenhouse gas (GHG) emissions, biowaste valorization, and so on [3]. In this sense, a serious challenge is avoiding soil fertility loss and increasing organic matter content, thus improving nutrient concentration, meeting higher agricultural production expectations, and conserving soil biodiversity [3,4]. As a fundament of agricultural production, the soil is an automated and regenerating system, but it is becoming exhausted due to the intensive plant production and use of chemicals. An environmentally friendly solution to overcome most of the abovementioned shortcomings of modern agriculture to promote grain crop production and sustainability is the application of soil organic amendments.

Incorporating organic amendments in the soil dramatically improves its nutrient status and water-holding capacity because it carries an enormous negative charge [3,5,6]. As an organic product resulting from the recycling of biowastes, vermicompost is a cost-effective, symbiotic, bio-oxidative process of interaction between earthworms and the composting microbiome that biochemically degrades biowastes and presents many properties to improve soil health and microbial diversity. It has vast potential to promote grain crop production, incrementing soil organic carbon with minimal environmental impact [7,8]. Researchers found that the incorporation of vermicompost was conducted to increase soil microbial biomass carbon and promote dehydrogenase, β-glucosidase, urease activities, and soil respiration as a whole [9]. The grain yield of barley was increased by 66% until five folds after soil incorporation of vermicompost in concentrations from 25 to 250 t ha^−1^. It also supported the soil organic and nutrient element content [10]. In addition, Khalifa and coworkers found the vermicompost to be a beneficial organic amendment that improves physical and chemical soil properties, especially soil organic carbon and available nitrogen [11,12]. It also positively influenced microbial biomass, dehydrogenase, and phosphatase soil enzyme activity [13,14].

Biochar is the product of biomass pyrolysis that represents a carbon-rich substrate useful to sequester soil carbon from soil and improve soil fertility [11,15,16]. During the last decade, biochar application has been accepted as a tool to enhance soil nutrients and quality and to reduce GHG emissions [17,18]. It may also retain up to five times its weight of water and nutrients such as mineral nitrogen [19,20].

Compost and biochar effectively improve soil nutrients and stimulate soil microbiome development. They provide substrates for microorganisms, thus altering soil microbial communities and the microbiome in different ways, including enzyme activities and cell replication, microbiome–plant interactions, microbial extracellular compound formation, and so on [21,22]. Furthermore, degrading the organic matter results in nutrient supply to plants. According to Foster and collaborators, biochar significantly increases α-1,4-glucosidase, β-D-cellobiohydrolase, and β-1,4-N-acetylglucosaminidase, significantly decreases β-1,4-glucosidase and phosphatase activities, and slightly decreases microbial biomass C [23]. Compost and biochar are widely applied as organic fertilizers and soil structure improvers to stimulate plant growth [24]. In comparison, mineral fertilizers have different effects on soil microorganisms, which could be generalized as increasing microbial biomass dependent on pH values [25]. Thus, mineral fertilization modifies soil microbial community composition, stimulating the development of chemolithotrophic populations, but, in general, their application could reduce population number and diversity [26,27].

Soil microbial communities’ structure differs depending on environmental, nutritional, and agricultural factors [28]. The substrate utilization patterns of soil microbes are characteristic and reflect changes in the soil. In diverse studies, authors have reported the importance of amendment introduction to the soil for community structure and diversity [29,30,31]. Changes in C-source utilization under different fertilization levels could be observed by applying Ecoplates of the Biolog system. Thus, biochar could promote microbial communities’ polymer and carboxilic acid utilization capacity [32], while compost and *Pseudomonas* inoculants can increase amino acids and polymers utilization patterns [31,33].

Other researchers observed the controversial effects of applying organic amendments on the organic matter and nutrient elements accumulated in soil and plants [5]. In addition, our literature survey found that little scientific research has been conducted recently on applying vermicompost and biochar in feed barley fields. We hypothesized that organic fertilization would positively affect the soil and microbial properties and, hence, grain yield and nutrient content. From this starting point, the objective of the present study is to explore the response of feed barley plants to the incorporation of organic and mineral fertilizers in soil and its effect on soil and microbial properties.

## 2. Materials and Methods

### 2.1. Field Description

The field experiment was carried out on the training and experimental field of the Agricultural University of Plovdiv with alluvial-meadow soil (42°08′15.4″ N 24°48′16.4″ E). Geographically, the site is located in the Thracian-Strandzha region. The alluvial-meadow soils are formed on sandy-loam and sandy-gravel quaternary deposits. According to the International Classification of FAO, it is Mollic fluvisol. They are formed on alluvial deposits and have a well-formed, humus-accumulative horizon, which gradually passes into the carbon horizon. A gleization process is observed deeply down (below 100 cm) in the soil-forming material—the A-C-G profile.

### 2.2. Experimental Design and Soil Amendments

The field experiment was performed in five treatments with four replicates per treatment. All twenty plots were prepared using randomized complete block design with a surface of 35 m^2^ (8.75 × 4 m) per plot. Winter feed barley (*Hordeum vulgare* L., *Zemela cultivar*) was sown at the end of October 2022 at 160 kg/ha seed density. The climatic conditions during the experiment could be characterized as average compared to the average values of multi-year data. The temperatures did not show significant deviations from those necessary for crop development. The average amount of precipitation during November and December of 2022 was 75 mm, while from January to April 2023, it was 138 mm, with a lower amount in March.

The treatments were set as follows: (1) untreated control (without amendments); (2) mineral fertilizer (MF, 100 kg/ha, NPK, 15:15:15); (3) vermicompost (VC, 12 t/ha); (4) vermicompost + mineral fertilizer (VC+MF, 6 t/ha + 50 kg/ha); and (5) biochar (BCh, 10 t/ha). The corresponding amendments were incorporated into the soil before using a soil cultivator to sow the barley seeds.

At the tillering phase, 50 kg/ha of nitrogen as ammonium nitrate (NH_4_NO_3_, 34.4% N) was applied in the mineral fertilization treatment as a part of the fertilization plant for barley. Agropolyhim AD, Devnya, Bulgaria, produced these commercial mineral fertilizers.

The vermicompost was produced from cow manure and wheat straw in proportions of 72.73%:27.27% (w:w). The process consisted of two stages—composting, carried out for three months, followed by vermicomposting of the resulting fresh compost for two months using the earthworms Eisenia fetida and Lumbricus rubelis [34]. The final vermicompost was sieved through 5 mm sieve. The process was carried out at the Bulver EOOD site in Kalekovets village, Plovdiv district.

Biochar from oats was produced at Agrosystemy Sp. z o.o. in Puławy, in the Lublin Voivodeship. The feedstock for pyrolysis was oats (grain), a plant rich in silicon. The prepared material was subjected to pyrolysis in a drum chamber at a temperature of 600 °C. Afterward, the entire batch was cooled, transported to the Experimental Station in Skierniewice, and added by weight to the compost pile. The entire pile was mixed every ten days using a vertical air aerator. The biochar was kindly supplied to the experimental field in Plovdiv by the Polish partner in ConnectFarms project—SGGW (Warsaw University of Life Sciences, Institute of Agriculture, Warszawa, Poland).

Plant protection activities were not performed to avoid the applied pesticides’ indirect effects on soil microorganisms. The water supply was only based on soil water content and precipitation.

The physicochemical analysis of vermicompost is described in detail in the work of Popova and collaborators [34]. The biochar analysis was performed as described in the study of Jindo and coworkers [35]. Total nitrogen was determined using the Kjeldahl method, while the total carbon content was measured with a CHNS Vario Cube Macro analyzer. In brief, 2 g of biochar was combusted in a muffle furnace with the following temperature program: initial temperature: 200 °C for 2 h; second step: 400 °C for 1 h; and final temperature: 550 °C for 2 h. After cooling, 5 mL of 10% HCl was added and the mixture was evaporated to dryness at 120 °C. The sample was then moistened with deionized water and 10 mL 2 M HNO_3_ was added. The solution was quantitatively transferred to a 50 mL flask. After the quantitative transfer, the solution was filtered. The phosphorus content was determined using the vanadomolybdate method (ISO 6878:2004). The remaining elements were determined using the ICP method.

The analytic data showed higher concentrations of most nutrient elements as total content (nitrogen, potassium, calcium, and magnesium) in biochar (Table 1). Only phosphorus and iron were found to be higher in vermicompost.

Table 2 shows the main macronutrients in the corresponding amendments added to the soil per treatment.

### 2.3. Soil and Plant Analyses

The analyses of soil and plant samples were performed using widely applied methods.

#### 2.3.1. Soil Analyses

The soil samples were taken from the barley rhizosphere twice—(1) before the application of amendments and sowing of seeds and (2) during the fruit development of barley (BBCH 75: medium to late milk). The sampling was carried out from each replication randomly, followed by air drying and sieving (2 mm).

For the determination of pH and EC, the samples were dispersed in deionized water (1:5, *w*/*v*) and shaken in a reciprocal shaker for 30 min, followed by sedimentation for 10 min and measurement with laboratory pH—EC—meter [36,37].

The accessible soil nitrogen (N-NH_4_ and N-NO_3_) was extracted with 1% KCl. Twenty grams of fresh soil equivalent of dry mass was placed in a bottle with 100 mL of 1% KCl. After 1 h shaking at 120 rpm, the resulting slurry was filtered. Fifty milliliters of the filtrate was mixed with 10 mL of 40% NaON in a distillation flask of the Parnassus—Wagner apparatus and 5 mL B(OH)_3_. After the distillation, 50 mL of distillate was mixed with 25 mL of 10% FeSO_4_ and 5 mL of 0.5% Ag_2_SO_4_ as a catalyzer. This was followed by distillation in another receiver flask with 10 mL 4% H_3_BO_3_ and 1–2 drops of mixed indicator. The first distillate with N-NH_4_ and the second with N-NO_3_ were titrated with 0.02 N H_2_SO_4_ until the blue colored solution acquired a faint pink color. A blank sample, only KCl, was carried out in parallel [38,39].

To extract available forms of P, 0.04 N Ca-lactate buffered with HCl was used. Two grams of soil were poured with 100 mL 0.04 N Ca-lactate solution, shaken for 90 min at 120 rpm, and filtered. Ten milliliters of the filtrate was placed in a flask of 100 mL before adding 10 mL 0.1 N N_2_SO_4_ (p.a., 96–97%, Fluka, Buchs, Switzerland), 10 mL of ammonium molybdate solution (p.a., 99%, Fluka), and deionized water until a total volume of 100 mL; finally, 5 drops of SnCl_2_ (p.a. 98%, Sigma Aldrich, St. Louis, MO, USA) was added as a reductant. The sample was allowed to stand for 12 min to produce the color reaction (blue coloration). The solution optical density was determined at 700 nm. The quanification of P_2_O_5_ was determined according to a previously prepared standard curve (colorimetric method). [40].

Mobile potassium was extracted from 1 g of dried soil by adding 25 mL 1 N HCl. After overnight shaking (24 h) at room temperature (20 °C), the soil sample was filtered (Whatman No. 1) and rinsed to 100 mL with deionized water. The determination was performed with a calibration curve, standard solutions, and a PFP-7 flame photometer (Jenway, London, UK) [41].

The soil’s total organic carbon pool was assessed using the potassium dichromate method. A 0.4 g soil sample was mixed with 5 mL of dichromate solution and 7.5 mL of sulfuric acid. The mixture was heated at 135 °C for 30 min and rapidly cooled to room temperature. Fifty milliliters of distilled H_2_O was added, rapidly cooled, and raised to 100 mL. After one hour of settling, the suspension was centrifuged for 10 min at 2000× *g*. The final determination was conducted with a spectrophotometer at 585 nm with a standard curve of D-glucose.

#### 2.3.2. Plant Tissue Analyses

After harvest, 10 g of plant seeds of each replication plot were dried for 96 h at 60 °C. They were milled and prepared for chemical analyses. A total of 0.5 g of milled grains were digested with concentrated H_2_SO_4_ (p.a., 96–97%, Fluka) and H_2_O_2_ oxidant (wet mineralization) in a digester at 420 °C until the total sample mineralization (crystal clear). After the mineralization, the digested plant material was transferred into 250 mL volumetric flasks and raised with deionized water to the mark. The total nitrogen content was determined using the Kjeldahl method in the Parnassuss–Wagner apparatus. The procedure is described in Section 2.3.1 [42]. The total phosphorus content was assessed calorimetrically (2 mL from digested plant material was transferred in 100 mL a volumetric flask followed by 10 mL 0.1 N H_2_SO_4_ (p.a., 96–97%, Fluka); 10 mL 2% ammonium molybdate solution (p.a., 99%, Fluka); and reductant SnCl_2_ (p.a. 98%, Sigma Aldrich) using a Camspec M105 spectrophotometer at a wavelength of 700 nm, while the total potassium content was estimated by directly measuring the digested and filtered solution using a PFP-7 flame photomter. The grain protein content was calculated from the total N content using a conversion factor of 5.83, according to FAO [43].

### 2.4. Soil Microbial Analyses

One soil sample (consisting of three sub-samples from each replicate) for microbiological analyses was taken from the barley rhizosphere and stored in the fridge for several days before analyses.

#### 2.4.1. Basal and Induced Soil Respiration

Basal soil respiration (BR) was carried out using the method described by Alef [44]. It consisted of placing moist soil (50 g of dry soil equivalent) with an adjusted water holding capacity (WHC) of 55% in a closed container with a small beaker with 20 mL 0.05 M NaOH at 22 °C. After the desired exposure time (usually 24 h), the reaction was stopped, and 1 mL of BaCl_2_ was added to the beaker to precipitate the carbonates. In continuation, two drops of phenolphthalein were added as a color indicator of acidity. The residual NaOH was titrated with 0.05 M HCl until it changed from violet to colorless. To estimate the weight of atmospheric CO_2_, a control sample without soil was used, and 1 mL of 0.05 M of NaOH was consumed, equivalent to 1.1 mg CO_2_. All the samples were elaborated in triplicate. The quantification of CO_2_ production was calculated according to the following formula:CO_2_ (mg/SW/h) = (V_o_ − V) × 1.1/dwt × 24(1)
where SW is the amount of soil dry weight (g); Vo is the volume of HCl used for titration of the blanks (mL); V is the volume of HCl used for titration of the sample (mL); dwt is the dry weight of 1 g of fresh soil; 1.1 is the conversion factor (1 mL 0.05 NaOH equals 1.1 mg CO_2_); and 24 is the incubation time in hours.

In continuation, the induced soil respiration (SIR) was studied for six hours to obtain the maximal initial respiratory response of the microbial biomass when applying a carbon source at 10,000 mg kg^−1^ C as glucose to the soil, following the same procedure as per basal respiration. The soil microbial biomass C (SMB) was estimated using the correlation of SIR and the fumigation–incubation method described by Anderson and Domsch [45]. The metabolic quotient was calculated as a ratio of basal respiration and microbial biomass. For that purpose, the metabolic quotient (MQ) was calculated as a ratio of BR to SMB.

#### 2.4.2. Soil Enzyme Activities

The dehydrogenase enzymatic activity was estimated through the method of Thalmann [46] modified by Alef [47]. It consisted of reducing triphenyl tetrazolium chloride (TTC) to thriphenyl formazan (TPF) by microorganisms. In brief, 5 g of moist soil was mixed with 5 mL 0.35% TTC solution in test tubes. After incubation for 24 h at 30 °C, 40 mL of acetone was added, mixed, and incubated for two hours in the dark. After filtration, estimation of the quantity of TPF was performed spectrophotometrically at 546 nm.

The phosphatase activity was measured as described by Eivazi and Tabatabai [48], using *p*-nitrophenyl phosphate (*p*NPP) as a substrate. In brief, 1 g moist soil with 55% of WHC was placed in a flask with 0.25 mL of toluene, 4 mL of buffer, and 1 mL of 15 mM *p*NPP solution. The mixture was incubated at 37 °C for one hour. After that, 1 mL 0.5 M CaCl_2_ and 4 mL 0.5 M NaOH were added and mixed, followed by filtering and measuring the absorbance at 400 nm on a spectrophotometer.

In addition, the β-glucosidase activity was assessed using the method of Tabatabai [49] modified by Alef and Nannipieri [50]. The method consisted of *p*-nitrophenyl-β-D-glucoside (PNG) degraded by microorganisms to *p*-nitrophenol. In brief, 1 g moist soil equivalent with 55% WHC was placed in a flask with 0.25 mL of toluene, 4 mL of buffer, and 1 mL of PNG solution. The mixture was incubated at 37 °C for one hour. After that, 1 mL of CaCl_2_ and 4 mL of Tris buffer, pH 12, were added, mixed, and filtered. The absorbance was measured at 400 nm on a spectrophotometer, and the final quantity of p-nitrophenol was calculated using a standard calibration curve.

### 2.5. Community-Level Physiological Profiling with Biolog^TM^ Ecoplates

The microbial rhizosphere communities of barley were assessed through Ecoplate, Biolog Inc., Hayward, CA, USA. The plate had 96 wells in three replicates—31 unique C-sources and 1 control well. Ten grams of fresh rhizosphere soil were dispersed in 90 mL of distilled water with 0.85% NaCl, stirred for 20 min at 250 rpm on a rotary shaker. After settling for 15 min, the samples were diluted ten times, twice over. The inoculation of Ecoplates was achieved by adding 150 μL through a multichannel pipettor followed by incubation at 25 °C. The wells’ absorbance was assessed on a Biolog^TM^ MicroStation^®^ reader at 590 nm and 750 nm wavelength on 24 h basis for 7 days. The obtained data were normalized [51], and the carbon sources were grouped by their nature as amino acids (AAs), amines and amides (As), polymers (Ps), carbohydrates (CHs), carboxylic acids (CAs), and phenolic compounds (PCs) [52].

The average well color development (AWCD) was determined through the next formula:AWCD = ∑(A − B)/n
where A is the optical density (OD) in the well, B is the blank well reading, and *n* is the total number of carbon sources. The AWCD showed the community’s ability and resilience through the diverse carbon-source utilization capacity.

In continuation, we used Shannon (H’) and Pielou (E) indexes to estimate the community homogeneity description. The Shannon diversity index was calculated as follows:H′ = −∑[(n/N) × (ln *n*/N)]
where H′ is the Shannon diversity index, while *n* is the number of wells with a positive reading (OD > 0.25) to the total number of C-sources (N).

The Pielou evenness index (E) was calculated from the Shannon index and represents a ratio of H’ to the value of the substrate richness (S, number of wells with OD over 0.25):E = H′/ln S;

### 2.6. Statistical Analysis

All analyses were carried out in four replicates. One-way analysis of variance (ANOVA) using SPSS Statistics v.18 was carried out for variables at a probability level of *p* ≤ 0.05, as was principal component analysis (PCA). The significance of differences between mean values of the studied parameters was assessed through the least significant difference (LSD) test. Results are presented after calculating the mean and standard errors using Microsoft Excel v.14.

## 3. Results

### 3.1. Soil Properties

The soil analyses before the study showed low TOC—1.15% (Table 3). The soil solution reaction was slightly alkaline, while the electrical conductivity was relatively low. Based on the generally accepted limit values for the content of macroelements in soil, it was found that it was poorly supplied with accessible nitrogen and very well supplied with mobile phosphorus and potassium.

### 3.2. Barley Yield and Accumulation of Nutrients

The barley yield was affected by the incorporation of the amendments (Figure 1A). Yield quantity was highest in the vermicompost treatment by 20.1% compared to the untreated control and 15.9% compared to mineral fertilization treatment. The application of MF increased the yield by only 3.6% compared to the control and by 2% regarding the treatment with biochar incorporation. The combination of vermicompost and mineral fertilizer application at half concentration in the fourth treatment resulted in increased barley grain yield compared to the control by 10.8% and compared to MF by 7%. It seems that the organic fertilizer supplied different nutrients to the mineral fertilizer, but the yield was still lower than the application of vermicompost alone by 7.7%. Biochar incorporation was conducted to 1.6% increments of the grain yield compared to the control.

The application of amendments resulted in a difference in plant density per square meter during barley development. In our study, the number of plants differed depending on the soil amendments (Figure 1B), but the order was very similar to that of grain yield. The highest number of plants per square meter was found in the treatment with vermicompost incorporation, while the lowest was in the case of untreated control. The combined treatment was second in the number of plants, with 5.7% less compared to the VC one. No statistically proven difference was found between MF and biochar treatments. The number of barley plants in those amended treatments was 9.2–12.7% higher than in the control.

The nutrient content followed the yield tendency with higher grain N content and protein in the treatment with vermicompost application, which was 7% higher than the control and 3.8% compared to that with mineral fertilizer (Table 4). Only the data on vermicompost treatment were statistically significant compared to the rest. Despite the existing differences in grain phosphorus content, they were not significantly different from each other. However, the potassium content in grains differed between the treatments. The highest concentrations were found in treatments with a single application of organic (vermicompost) or mineral fertilizer. It was considerably higher than the rest of the treatments—about 19% compared to the control, 10% compared to the combined MF+VC application, and 14% compared to biochar incorporation.

### 3.3. Soil and Microbial Properties

The application of amendments influenced changes in soil and microbial properties in the upper soil horizon (Table 5). The TOC was increased in treatment VC by 70.2% compared to the control, 45.9% compared to the MF, 27.7% compared to the BCh, and 18.2% compared to the MF+VC treatment. According to this parameter, a statistically proven difference was found only in the case of vermicompost application.

The amendments applied to the soil also influenced changes in main nutrient content (mobile N, P, and K; Table 5). The introduction of mineral fertilizer had the most substantial impact on the mobile nitrogen concentration in soil, reaching 2.39 mg 100 g^−1^ of soil, followed by vermicompost with 2.11 mg 100 g^−1^ and the combined treatment with 2.04 mg 100 g^−1^. The lowest concentration was found in the unamended control. Similarly, the mobile phosphorus content in the studied soils was highest in VC treatment with significant differences with the rest—48.6% higher than the control, 23.3% than the biochar treatment, and 39.6% than in the treatment with combined application of MF and VC. The differences among treatments were smaller in the case of mobile potassium soil concentrations. Thus, the concentration in all amended treatments was higher and statistically proven compared to the control, in the case of potassium.

The amendments’ application affected the soil’s physicochemical properties (Table 6). The soil pH in the rhizosphere was reduced from 8.2 to less than 8. All amended treatments showed values higher than the control and statistical significance.

The incorporation of amendments influenced soil conductivity. This was clearly pronounced in the case of vermicompost incorporation, with a 12.5% increase compared to the control. Concerning the control, higher EC was also found in the MF treatment, at 2.95%. The lowest statistically proven EC was denoted in the treatment with biochar application.

The application of amendments, especially vermicompost treatment, influenced the microbial properties. A statistically proven increment was found in both cases of basal soil respiration and substrate-induced respiration. BR in the treatment with VC was increased by 52.4% compared to the control, 46.5% compared to the MF, and 35.4% compared to the BCh treatment. Respectively, SIR in the VC treatment showed a lower increase regarding the BR. The rate of CO_2_-C production of the soil active microbiome in VC treatment was 32.1% higher compared to the control and 24% compared to the combined treatment. Thus, the microbial quotient after vermicompost application differed considerably from the rest of the treatments.

Soil enzymes were influenced by the amendments’ incorporation (Table 7). The dehydrogenases were more active in the VC treatment compared to the control and the biochar treatments. Similarly, β-glucosidase enzyme activity was much higher in the VC treatment—29.4% to the control, 37.5% to the MF, and 24.53% to the MF+VC. A higher difference was observed compared to the biochar treatment, by 43.48%. In the soil phosphatase case, higher activity was found in the MF treatment concerning the control, biochar, and combined treatment.

### 3.4. Carbon Sources Utilization Capacity of Soil Microbial Communities

The AWCD showed different metabolic rates of carbon-source utilization during the experiment (Figure 2). From the beginning, the slope of the curves was steep at all time points until the end. The community utilization rate was lower in the VC treatment during the whole study. After 144 h, the utilization of VC and BCh was similar, while that of the rest of the treatments was slightly higher. The highest AWCD value was observed in MF, followed by the control and MF+VC treatments.

The 31 carbon sources of Ecoplates are separated based on their nature into six groups. The utilization differed among the groups, as well as among the treatments (Figure 3). Higher utilization rates were found for amino acids, polymers, and carboxylic acids, while lower values were found for carbohydrates. The lowest consumption was observed in wells with phenolic compounds, amines, and amides. We found specific differences in groups of C-source utilization patterns between treatments. Higher ability to use amino acids as a C-source was found in MF and MF+VC treatments. Mineral fertilizer also promotes a higher consumption of carbohydrates and phenolic compounds. Lower utilization patterns were observed in the case of carbohydrates, amines, and amides in soils amended with VC. Higher consumption of polymers was found in the control and treatment with biochar.

Figure 4 shows the AWCD utilization capacity of each one of the carbon sources in corresponding treatments. The most complete utilization source was the itaconic acid in MF, with a difference of 30 to 37% compared to the rest of the treatments. D-mannitol (MF) and γ-hydroxybutyric acid (VC) were also C-sources with high utilization rates. Still, the difference with the rest of the treatments was not as high as in itaconic acid (8–17% and 1–20%). Lower utilization was found for 2-hydroxy benzoic acid, α-ketobutyric acid, and D,L-α-glycerol phosphate. The last one had good utilization in VC treatment but was very low in the rest.

We used 120 h data to calculate barley rhizosphere microbial community diversity indexes (Table 8). Despite amendments incorporated into the soil before barley sowing, the diversity indexes did not differ significantly between treatments. The diversity, according the Shannon–Wiever index, was highest in the MF and BCh treatments. Lower data were observed in the treatment with VC addition. The difference between these treatments was statistically proven. Intermediate data were found for the diversity in the MF and MF+VC treatments. The results of the Pielou index and richness in the treatments were not statistically different.

Figure 5 represents a biplot in the first and second principal component (PC) dimensions under different fertilization cases. The carbohydrates and polymers had positive values for the first and second PC. The highest absorption rate of any treatment in the experiment distinguished these two carbon groups. PC1 contributed 90.6% to the total variation and correlated greatly with amino acids, carboxylic acids, and polymers. PC2 contributed 7.5% and was associated mainly with amines and amides. The lowest absorption rate had phenolic compounds, amines, and amides, where even the amines and amides had negative values for PC1, while the phenolic compounds had negative values for both PCs. The highest correlation between the control and BCh was found concerning C-source consumption. As can be seen from the graph, they form an acute angle. The lowest correlation of the control is found with VC, followed by the MF and combined MF+VC treatments.

In Figure 6, the C-source utilization in treatments is compared using PCA. In this case, both PCs formed 94.2% of the total variation. PC1 influenced 74.9% of the variations, while PC2 had 9.3% weight. The main possitive association of PC1 was with E3, D2, B4, H1, and D4 (γ-hydroxybutyric acid, D-mannitol, L-asparagine, α-D-lactose, and L-serine, respectively). The lowest association with this component was observed for C3, H2, G3, H4, and G2 (2-hydroxy benzoic acid, D,L-α-glycerol phosphate, α-ketobutyric acid, putrescine, and glucose-1-phosphate, respectively). The highest positive rate of influence of PC2 was found for H2, H3, and B2 (D,L-α-glycerol phosphate, D-malic acid, and D-xylose, respectively), while the lowest was for B1, C2, G4, F4, and F3 C-sources (pyruvic acid methyl ester, i-erythritol, phenylethylamine, glycyl-L-glutamic acid, and itaconic acid, respectively). The highest correlation of the control was found with VC. As can be seen from the graph, they form an acute angle between themselves. The treatments MF and MF+VC formed the highest correlation with each other, as seen from the graph; they form an acute angle among themselves.

## 4. Discussion

Adding amendments to the soil alters soil properties and, thus, may positively influencing plant growth, development, and yield depending on the conditions, soil properties, kind of amendments, and so on [7,10]. Nutrient deficiency and stresses, such as water scarcity and salinity, lower plant growth, yield, and soil microbiome activity [22]. Crop growth, the development of soil microorganisms, and nutrient availability are highly dependent on soil response. At the same time, the pH in the control was reduced, which is attributed to organic matter degradation processes taking place in the experimental soils. This further brings to the fore the increased pH values in the amended treatments. They could be related to the difference in pH of added substrates or the contribution of soil microbial activity to the immobilization of nutrients in microbial biomass. Other researchers found similar results concerning soil reaction [9,53,54,55]. This was more pronounced in acidic rather than low-alkaline soils [54]. Other investigators did not find statistically proven differences when adding organic or mineral fertilizers [55].

Higher soil respiration rates result mainly from the microbial degradation of soil organic compounds [9,56]. Compared to the unamended control and traditional mineral fertilization, we tested the influence of organic amendments (vermicompost and biochar) on the number of plants of barley, grain yield, soil, and microbial properties. We found significantly higher barley parameters (plant number, grain nutrients, and grain yield) in the treatment with vermicompost incorporation. The amendments’ application changed the soil and microbial properties. TOC and whole microbial activity were significantly increased in the presence of the vermicompost. The application of vermicompost influenced the development of strong β-glucosidase- and dehydrogenase-producing microbial communities.

Different researchers emphasized the importance of organic application to the soil, especially the application of vermicompost, which promoted soil health and nutrient availability, thus improving plant productivity [12]. According to the studies, the benefits to plants derive from the nutrients available in the vermicompost and the microbial communities introduced with it [57]. Such studies confirmed our findings that the vermicompost boosts enzyme activity, especially β-glucosidase and acid phosphatase, and also increases soil microbial biomass C [58,59]. As fewer research studies have been published on vermicompost application in barley [60], our data from the present study resulted in even more valuable results. We also found that Khalifa and coworkers reported maximum barley yield and microbial activity in the case of rice straw and animal waste vermicompost incorporation in soil [12,61]. In this way, Shen et al. found improved barley grain yield after increasing doses of digested sewage sludge vermicompost application to the soil [10]. Other results showed enhanced soil organic carbon, N, and P concentrations after vermicompost incorporation. The observations of Valdez-Pérez et al. are also consistent with our results [62].

We found higher MQ in compost-amended soil, followed by the biochar treatment. Like our study, Hu and coworkers observed a significant increase in soil metabolic functions after organic matter addition [63]. These results differ from those reported by Zhou and collaborators, where biochar application was conducted to reduce the quotient in a diverse way concerning the control [18]. They attributed this effect to the kind of biochar, temperature of pyrolysis, pH, and the soil’s organic content. Other researchers discussed that the MQ rate could also depend on N availability and liability in the soils [64,65].

The characterization of physiological profiles of microbial communities contributes to the overall description of microbiome changes as a consequence of amendment application. The characterization of physiological profiles of microbial communities contributes to the overall description of microbiome changes as a consequence of amendments application. We observed an outstanding response from the communities to the application expressed as C-source utilization (Figure 2). It was highly pronounced for amino acids and carboxylic acids in treatments with MF and MF+VC, while using polymers was associated with the control (Figure 5). Other researchers found higher AWCD after diverse organic and mineral fertilization than the control treatment (no amendment application) [66]. They related the functional diversity of microbial communities with the presence of D-cellobiose, glucose-1-phosphate, ß-methyl D-glucoside, methyl pyruvate, D-galactosate gamma lactone, D-mannitol, N-acetyl-D-glucosamine, D-galactosalonic acid, and L-serine, which is partially covered by our results. They reported that combined application of organic and inorganic amendment (MF+VC in our study) increased the Shannon–Wiener index but decreased the Pielou evenness index, which does not follow our results. We found increased indexes, like Yu and co-workers did [67]. We supposed that the inorganic amendment could supply fast nutrients for plants in addition to the vermicompost, which increased soil C-sources.

Composts and biochars are very different, and their properties and effects on barley plants depend on many factors, including the type of soil and the biowaste from which they originate, among others [68]. Researchers reported that biochar derived from sheep manure significantly increased barley yield, but vermicompost had no such effect [69]. In another study, biochar application (13.5, 20.25, and 27 t/ha) increased barley grain yield from 3% to almost 20%, which corresponds to our findings [70]. The stimulating effects of biochar on soil microbial biomass and activities, as well as the increment of organic carbon content in soil, were observed by Nasiri and coworkers [11]. In such research, they found increased microbial biomass after biochar application, corresponding to our findings [corresponding to MQ] and those of other researchers [71,72]. They hypothesized that this was a prerequisite for the increased grain yield they found. Rekaby et al. also found significantly increased plant growth after incorporating plant residues, compost, and biochar into the soil [53], where the biochar effect was higher than that of the compost.

The functional diversity of rhizosphere microbial communities shows their ability ro utilize as many C-sources as possible. This show the ecological resilience of the communities.

It seams that mineral fertilization promotes the amino acid and carboxylic acid physiological profiles in these communities (Figure 3).

## 5. Conclusions

In the present study, we found improved grain yield and number of plants when amendments were applied. The best results were found in the case of vermicompost in combination with mineral fertilizer. The vermicompost addition resulted in improved nitrogen grain content and soil organic carbon. A statistically proven increase in soil microbial respiration and enzyme activities, especially of the β-glucosidase, was also observed.

The results suggest a positive influence of the organic amendments, especially vermicompost, on plant growth, soil, and microbial characteristics, which should be developed in further investigation.

## Figures and Tables

**Figure 1 microorganisms-12-01447-f001:**
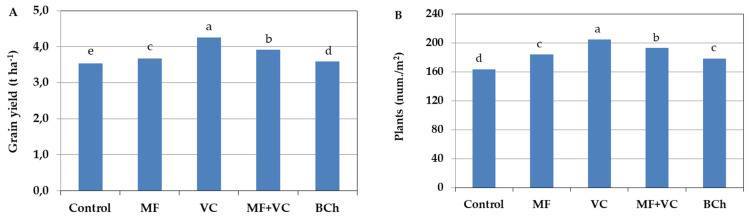
Barley grain yield affected by different fertilization types (**A**) and the number of plants per square meter (**B**). The results shows the mean of four replicates. Different letters indicate statistically significant differences between treatments at a significance of *p* ≤ 0.05. MF—mineral fertilizer; VC—vermicompost; MF+VC—combined application; BCh—biochar.

**Figure 2 microorganisms-12-01447-f002:**
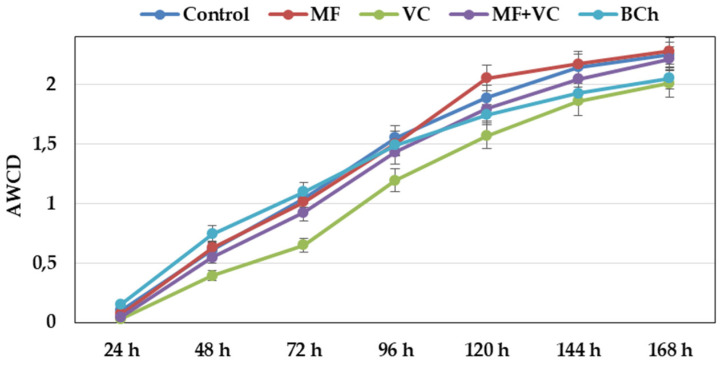
Changes in average well color development over time of soil microbial communities at 590 nm. Data represent the means and the standard errors (n = 3).

**Figure 3 microorganisms-12-01447-f003:**
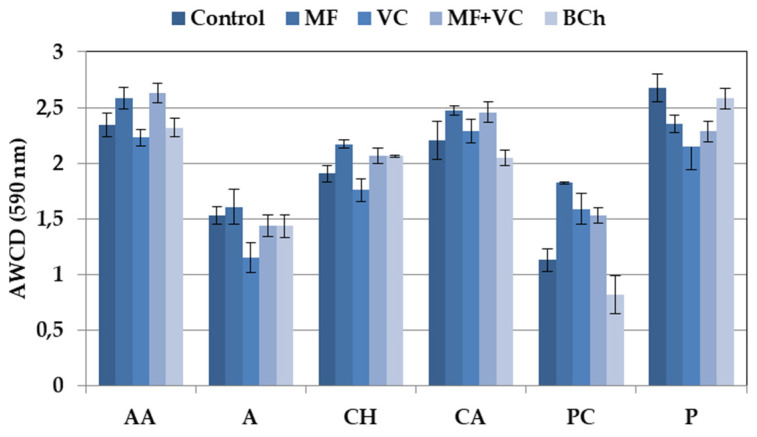
Average well color development of different groups of C-substrates. Data represent the means and the standard errors (n = 3).

**Figure 4 microorganisms-12-01447-f004:**
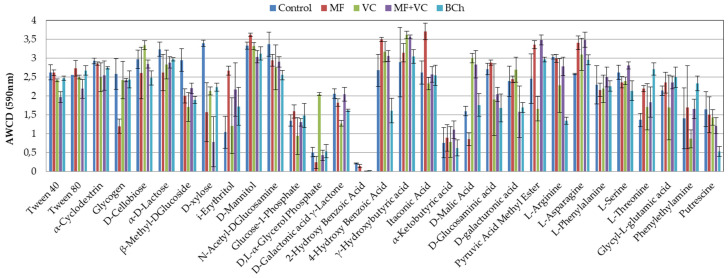
Average well color development of each one of the 31 substrates of the Ecoplates at 590 nm at the end of the study. The results show the means and the standard errors (n = 3).

**Figure 5 microorganisms-12-01447-f005:**
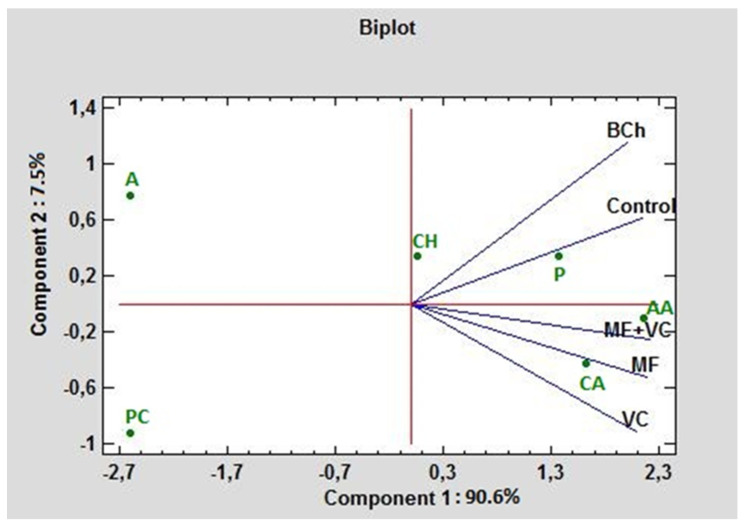
PCA ordinations for different carbon sources (indicated by AWCD) of ecoplates at 590 nm. AA—amino acids; A—amines and amides; CH—charbohydrtaes; CA—carboxylic acids; PC—phenolic compounds; P—polymers.

**Figure 6 microorganisms-12-01447-f006:**
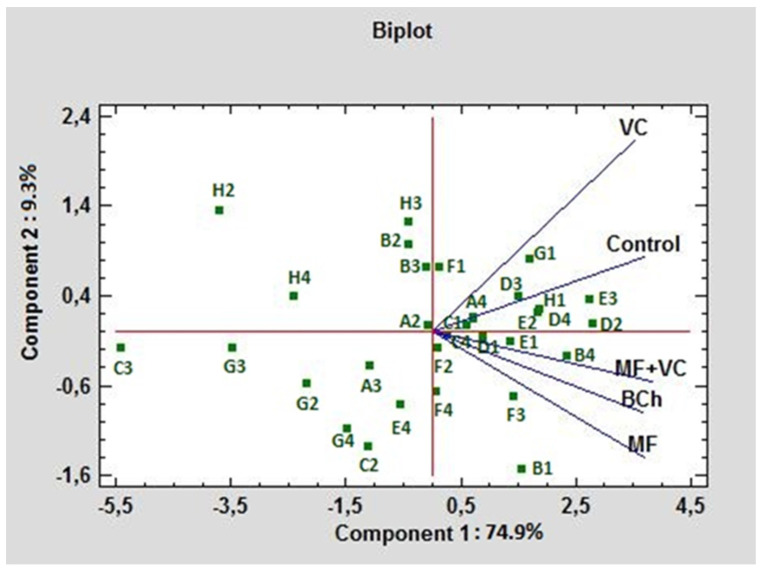
PCA ordinations for different carbon sources of ecoplates at 590 nm utilization under different fertilization treatments.

**Table 1 microorganisms-12-01447-t001:** Physicochemical properties of vermicompost and biochar on a dry weight basis. Data show mean and standard error (n = 3).

	Unit	Vermicompost	Biochar
**TOC**	(%)	18.09 ± 1.5	65.00
**pH**		7.8 ± 0.01	-
**EC**	(µS cm^−1^)	1690 ± 10.1	-
**Total N**	(%)	1.27 ± 0.18	1.31
**P**	(%)	1.09 ± 0.09	0.64
**K**	(%)	1.63 ± 0.21	7.95
**Ca**	(%)	3.12 ± 0.37	4.28
**Mg**	(%)	0.74 ± 0.08	3.06
**Fe**	(%)	0.53 ± 0.05	0.13

**Table 2 microorganisms-12-01447-t002:** Main macronutrients added in the soil. MF—mineral fertilizer; VC—vermicompost; MF+VC—combined application; BCh—biochar.

Treatments	N	P	K
	(g m^−2^)		
Control	0	0	0
MF	6.50	1.50	1.50
VC	15.24	13.08	19.56
MF+VC	10.87	7.29	10.53
BCh	13.1	6.4	79.5

**Table 3 microorganisms-12-01447-t003:** Physicochemical properties of the soil of the studied field before the application of amendments on a dry weight basis. Data show mean and standard error (n = 3).

	pH	EC (µS cm^−1^)	Accessible N (N-NH_4_+N-NO_3_) (mg 100 g^−1^)	P_2_O_5_ (mg 100 g^−1^)	K_2_O (mg 100 g^−1^)	TOC (%)
Agricultural field	8.2 ± 0.0	116.0 ± 1.9	1.49 ± 0.02	3.64 ± 0.30	9.7 ± 0.2	1.15 ± 0.10

**Table 4 microorganisms-12-01447-t004:** Protein and main nutrient content in barley grains. The results show the mean of four replicates. Different letters indicate statistically significant differences between treatments at a significance of *p* ≤ 0.05. MF—mineral fertilizer; VC—vermicompost; MF+VC—combined application; BCh—biochar.

	Protein (%)	N (%)	P (%)	K (%)
Control	6.83 b	1.17 b	0.05 a	0.70 b
MF	7.07 b	1.21 b	0.05 a	0.86 a
VC	7.44 a	1.28 a	0.07 a	0.86 a
MF+VC	6.98 b	1.20 b	0.06 a	0.77 b
BCh	6.87 b	1.18 b	0.06 a	0.74 b
LSD	0.3604	0.0618	0.0315	0.0506

**Table 5 microorganisms-12-01447-t005:** TOC, N, P, and K content of the soils in each treatment at the end of the experiment. The results show the mean of four replicates. Different letters indicate statistically significant differences between treatments at significance of *p* ≤ 0.05. MF—mineral fertilizer; VC—vermicompost; MF+VC—combined application; BCh—biochar.

	TOC (%)	Accesible N (N-NH_4_+N-NO_3_) (mg 100 g^−1^)	P_2_O_5_ (mg 100 g^−1^)	K_2_O (mg 100 g^−1^)
Control	0.84 b	1.88 b	1.85 d	2.93 b
MF	0.98 b	2.39 a	2.69 ab	3.44 a
VC	1.43 a	2.11 b	2.75 a	3.66 a
MF+VC	1.12 b	2.04 b	1.97 cd	3.68 a
BCh	1.12 b	1.89 b	2.23 bc	3.63 a
LSD	0.3106	0.2738	0.3510	0.4481

**Table 6 microorganisms-12-01447-t006:** Soil physicochemical and microbial properties of the soils in each treatment. The results show the mean of four replicates. Different letters indicate statistically significant differences between treatments at a significance of *p* ≤ 0.05. MF—mineral fertilizer; VC—vermicompost; MF+VC—combined application; BCh—biochar; BR—basal respiration; SIR—substrate-induced respiration.

	pH	EC	BR	SIR	MQ
		(μS cm^−1^)	(μg CO_2_-C g^−1^ h^−1^)		
Control	7.58 b	129 c	2.71 b	3.52 b	21.1
MF	7.98 a	132 b	2.82 b	3.55 b	21.7
VC	7.86 a	145 a	4.13 a	4.65 a	24.4
MF+VC	7.88 a	126 c	2.92 b	3.75 b	21.3
BCh	7.94 a	121 d	3.05 b	3.73 b	22.4
LSD	0.2506	3.9011	0.6821	0.5044	-

**Table 7 microorganisms-12-01447-t007:** Soil microbial enzyme activities in each treatment. The results show the mean of four replicates. Different letters indicate statistically significant differences between treatments at a significance of *p* ≤ 0.05. MF—mineral fertilizer; VC—vermicompost; MF+VC—combined application; BCh—biochar.

	Dehydrogenase (μg TPF g^−1^ h^−1^)	β-Glucosidase (μg pNPP g^−1^ h^−1^)	Phosphatase (μg pNPP g^−1^ h^−1^)
Control	0.310 b	0.511 b	0.326 c
MF	0.342 ab	0.483 b	0.568 a
VC	0.402 a	0.656 a	0.451 ab
MF+VC	0.342 ab	0.532 b	0.395 bc
BCh	0.284 b	0.458 b	0.357 bc
LSD	0.0542	0.1083	0.0481

**Table 8 microorganisms-12-01447-t008:** Microbial catabolic diversity indexes based on EcoPlate analysis. The results represent the means and the standard errors (*n* = 3). Different letters indicate statistically significant differences between treatments at a significance of *p* ≤ 0.05.

	Shannon	Pielou	Richness
**Control**	3.14 ± 0.06 ab	0.956 ± 0.020 a	27.0 ± 1.53 a
**MF**	3.25 ± 0.02 a	0.976 ± 0.002 a	28.0 ± 0.58 a
**VC**	3.13 ± 0.04 b	0.970 ± 0.003 a	25.3 ± 1.33 a
**MF+VC**	3.23 ± 0.01 ab	0.972 ± 0.003 a	28.0 ± 0.00 a
**BCh**	3.24 ± 0.02 a	0.980 ± 0.004 a	27.3 ± 0.33 a

## Data Availability

No applicable.

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
