# Peer review of "Fertilization Type Differentially Affects Barley Grain Yield and Nutrient Content, Soil and Microbial Properties"

_microorganisms, 2024, doi:10.3390/microorganisms12071447_

Round 1

Reviewer 1 Report

Comments and Suggestions for Authors

The manuscript microorganisms-3010945 describes the results of treatment with vermicompost, mineral fertilizers, their combinations, and biochar on the growth and productivity of barley and changes in the chemical composition and biochemical activity of the soil. The data presented in the manuscript is interesting and important to the development of sustainable agriculture. However, I think that before accepting this manuscript into the Microorganisms, the authors should improve the following points:

1. In the Introduction, a few sentences should be added about the changes in microorganisms when using mineral and organic fertilizers. After reading the manuscript, I had a generally positive opinion. But why was this manuscript submitted to the Microorganisms? In the information for the special issue “Soil Health and Plant-Microbiome-Bioeffectors Relationship in Sustainable Agriculture” I did not find any keywords suitable for this manuscript. The authors study the activity of enzymes in rhizosphere soil. But there are a lot of root exudates in the rhizosphere. Could the enzymes studied by the authors be partially of plant origin? Additionally, the keywords of the manuscript include “soil microbiome.” There are no direct studies or discussions of microbiomes in the manuscript. The authors should either add microbiome research or remove this keyword. Overall, I think that this manuscript (after corrections) would be a good fit for the journal "Agronomy".

2. The description of methods and results requires numerous additions and corrections.

Lines 19–20: “an untreated control” - It would be more helpful to specify the soil type here.

Lines 78–83, tables 1 and 2: Was the description of the soil characteristics made by the authors? If yes, then these are the results of the study, not the materials. Only analysis methods should be indicated here, and the results should be transferred to Section 3. If not, then provide links to a description of the soil characteristics.

Lines 95–96: “the end of October 2022.” The results of field experiments can vary greatly from year to year. Climatic conditions are one of the important factors. Provide here a brief general description of the climatic factors observed during the experiment.

Lines 97–99: What is the rationale for choosing the amount of fertilizer applied?

Line 134: What was the colorimetric method used to determine the total phosphorus content?

For Tables 3-6, it would be useful to provide LSD values. In the tables presented, it is not always obvious why different letters are used for averages with a small difference, while the same letters are used for averages with a large difference. For example, TOC and P2O5 are in Table 4, and Phosphatase is in Table 6.

The manuscript does not contain a description of the results of the MQ calculation (Table 5). If these results are not discussed, why are they shown in Table 5?

Discussion of the results is, in many cases, incorrect. For example, in the table, the average values of the variants are marked with the same letters; the text says that there are no statistically significant differences, but a comparison is made of how much one value is greater or less than the second. For what? If the differences are not significant, then the observed differences are due to experimental error. See lines 229–233; table 4 shows the values “0.84b” and “1.12b”, i.e., the differences are insignificant. In lines 232–233, this conclusion is written, but in lines 229–230, “Applying biochar also contributed to the total organic carbon in soil – 33.33%.” These are insignificant differences! Why discuss them? The same incorrect discussion is given in lines 240, 251-252, 276, 281, 301 and 336.

3. Discussion of the results obtained and their comparison with the results of other researchers.

Lines 295-296: The combined application of mineral fertilizer and vermicompost (line 296) did not have a significant effect on grain nutrients (line 295).

The authors point out that biofertilizers obtained from different sources can have different effects on soil and plants. I think that it will be useful to indicate in the discussion the source of vermicompost for the references discussed (lines 311-312, 312-313 and 326-327), as well as the rate of application of vermicompost (line 321).

Minor remarks:

1. Line 20: “Hordeum” would be the correct spelling.

2. line 94: 8 x 4 = 32. Why was the area 36 m2?

3. What was the K value in vermicompost (Table 2).

4. Line 121: What is a “soil sample”? How many grams?

5. Lines 125–126: What substance was used to construct the standard curve?

6. Line 198 and figure 1B. The text shows square meters; the figure shows cubic meters. Why are the values ​​on the scale "Plant" accurate to tenths?

7. line 229: “44.05%”? The values in Table 4 differ by 18%. What is the point of indicating changes in percentages accurate to hundredths?

8. Table 5: It is enough to indicate EC values in the form of whole numbers (without hundredths).

9. Table 6: Values must be reported to the nearest thousandth.

Author Response

Dear Reviewer,

Thank you for your valuable comments and suggestions. Most of them were addressed and in our opinion, the quality of the manuscript was improved.

Please find the specific answers in the attached file.

Kind regards,

Stefan Shilev

Reviewer 2 Report

Comments and Suggestions for Authors

The subject of the manuscript is framed within a topic, namely the use of alternative fertilisers to improve agricultural productivity and simultaneously mitigate the associated environmental sustainability problems, which is extremely actually and interesting. There is an extensive literature on the subject, particularly on the effects of organic soil amendments such as vermicompost and biochar on different types of cultivars. The topic is important and challenging, but the submitted manuscript is poorly written and does not address the issues developed with a clear and straightforward approach. Significant improvements need to be made to the entire manuscript.         The following shortcomings were noted and need to be corrected.

Title: I would suggest modifying it to make it more consistent with the content of the work by emphasising the comparison with different types of soil improvers.

Abstract: in general can be considered satisfying; however, in line 21"....block design ..." the authors must specify better whether CRD (complete randomised design) or RCB (randomised complete block). They must do the same in Materials and methods (Lines 93-94). It is also stated that "Barley growth.......were studied" but the parameters analysed do not include those related to plant growth.

Keywords: too many keywords; I suggest deleting "microbial enzymes" and "field experiment".

Introduction

The introduction is poorly written and needs further enrichment. The arguments are framed in a superficial and unconnected manner and are not well supported by the mentioned bibliography. In contrast to the title, very little space is reserved for the main topic of the study conducted, namely the effects of using vermicompost as a soil improver.  Even more space is reserved for the use of biochar. It would be appropriate to better emphasise the purpose of the work and the expected novelty with respect to current knowledge

Materials and Methods

Significant improvements must be undertaken in this section.

Lines 83-85: "...- 1.15" inappropriate way of reporting and the unit of measurement is missing. There is a discrepancy between the parameters mentioned in the text and those in Table 1: organic matter in the text while TOC in the table; mobile nitrogen in the text while Total N in the table. If the value in the table really refers to total nitrogen, the C/N ratio would be incredibly anomalous: (expressing Total N as %) 1.15/ 0.00149 = 772. It must be made explicit which parameters are really being analysed.

Line 90: the caption of Table 1should be corrected: : "Chemical and physicochemical properties...."; also why " ...both studied fields..." ?

Line 94: “...distribuited with a surface of 36 m2..". Does this measurement refer to the individual plot or to the whole field? In the second case, would each plot measure 1.8 m2 ? isn't that too small?

Line 97: "NPK at 15%" must be correct, the amounts of the three macronutrients must be indicated separately (maybe it is NPK 15:15:15?).

Lines 99-103: it is essential to describe the biochar and vermicompost production methods in more detail, as well as provide more information on the commercial chemical fertilisers used (composition, name, brand, manufacturer....). The comment on biochar is unclear. On what basis were the quantities of added soil amendment chosen? Do these doses correspond to equal amounts of macronutrients? What are the optimal standard parameters for this cultivar? Why was it not also considered necessary to carry out a treatment consisting of the only addition of organic compost (cow manure and wheat straw)?

Also better describe the fertilisation plan (timing?) the irrigation plan and the pest protection plan.

Finally, the description of how the analyses were carried out on the soil improvers used (replicates, methods, chemicals, brand, instruments (manufactors, model, type), etc.), references) is completely missing.

Lines 110-112: the sentence" The soil samples from the barley rhizosphere were analyzed before sowing and during the fruit development (medium to late milk) phase. " needs some clarification:

_ " fruit development (medium to late milk) phase”? identify the plant's growth phase according to the BBCH scale

_ the text mentions analyses done before sowing, but after or before the addition of soil improvers? Contrary to what is specified in lines 110-112, Table 4 shows the results of soil analyses taken only at the end of the experiment as is specified in the caption of the table. Please clarify.

Contrary to what is specified in lines 110-112, Table 4 shows the results of soil analyses taken only at the end of the experiment as is specified in the caption of the table. Please clarify.

Lines 116-118: the description of the soil accessible nitrogen analysis is incomplete and there are no references to it; it is still confusing (see comment at lines 83-85) which actual nitrogen form is searched in the soil samples and the insufficient description of the method given does not help: in table 1, Total N is mentioned while in table 4, N is mentioned

The cited reference on available phosphorus analysis is dated (1960) and cannot be found online: the authors must provide an accurate description and/or cite the appropriate, traceable bibliography from which the method can be understood.

Based on the information reported, the method used for the determination of mobile potassium is unclear: again, the authors must provide a valid description and/or bibliographic reference attesting to the efficacy and reliability of the chosen analytical procedures, accompanied by all due information (chemicals, purity, brand name, instrument manufacturers, etc.). The above applies to the whole of section 2.3, where this requirement is not provided.

Lines 120-126: the above applies; also note that the ISO 14235:1998 method was withdrawn by the International Organisation for Standardisation in 2021

Line 128: In "....plant seeds were dried for 96 h at 60 °C." does this mean the seeds of all plants grown in the four replicates for each treatment or a number of plants sampled?: please specify the sampling plan of the plant material. In line 130 H2O2 is not a catalyst. Also paragraph 2.3.2 does not provide all the necessary information as specified in the note to Lines 116-118.                                                                                                         Line 138-139: specify when and how many soil samples are sampled per replication and treatment                                                                                Line 141-157: also in 2.4.1, the description of the method is imprecise and incomplete:                                                                                                      _ it is not specified that the container where the reaction takes place must be closed;                                                                                                         _specify "desired exposure time," (line 144)                                                   _ BaCl2 in which container is added? in the medium or in the beaker with NaOH?                                                                                                             _ perhaps "The free NaOH.." means residual NaOH?                                _Line 150: "The quantification of CO2 production was calculated using a formula." which formula?                                                                     _correct "......at 10 000 mg.kg-1 C ": the number is missing the comma or the dot and after mg remove the dot.                                                            _in line 157 what is meant by "BR"? perhaps it is the abbreviated form of Basal Respiration? specify                                                                          Line 158-178: also in paragraph 2.4.2 the notes in Lines 116-118 and 141-157 apply:                                                                                                _ Line 159: correct "The activity of enzyme dehydrogenase.... " in " The dehydrogenase enzymatic activity..."                                                          Line 162: what is the concentration of TTC solution?                                   In Line 166 what is the most soil equivalent?                                                In Line 167 what is the buffer? what is the concentration of pNPP solution? In Line 175 what is the buffer? what is the concentration of the PNG solution?                                                                                                        In Line176 Tris is for?

Paragraph 2.5 should be rewritten better: it is wrong to write "...means comparing by one-way ANOVA..." . Similarly in all figure and table captions it is wrong to write "... LSD test of ANOVA... "and should be corrected

Results:

In section 3.1, the grain yield results between the different treatments are compared. While in the text the parameter tiller number is named, in the figure it is otherwise referred to as plant number. In fig 1A, the unit of measurement of plant number must be corrected to m2 instead of m3.            

In materials and methods (section 2.3.2), parameters such as P total and K total are mentioned, why are their values listed in table 3 as P2O5 and K2O?                                                                                                            Also in table 3, the protein content has evidently been calculated from N-Kjeldhal, but what factor was used? from the ratios it does not appear to be a constant number.

In paragraph 3.2, the sentence " ...Applying biochar also contributed to the total organic carbon in the soil - 33.33%" should be corrected: the verb form "contributed" is not appropriate and furthermore the hyphen followed by the number is not a correct way of reporting a value (this should be corrected in the various points where it is repeated).

In line 235 the authors should make explicit the forms containing N, P and K to which the values in table 4 refer: the poor description of the methods in section 2.3.1, as I have already pointed out, does not help to identify them.

In the caption of Table 5, the meaning of the abbreviation MQ is not made explicit. In the same table, moreover, the unit of measurement given for the BRT and SIR values does not seem correct: it is usually given as mg C-CO2 g-1dw, futhermore, why is it quoted per hectare?

How do the authors interpret the observation in lines 253-254?

The term "regarding" in line 257 does not seem appropriate.

What is the authors' interpretation of the statement in lines 265-273?

The meaning and role of the statement in line 274 is not clear.

Why are the enzyme activity data also reported per hectare in table 6? What is the significate?

What is the meaning of the sentence in lines 276-277 "Data were statistically significant in the case of control and biochar treatments”?                                                                                                   

Discussion

The statement with which the authors begin this section (lines 289-290) is questionable. From the overall results presented, it is not possible to associate effects determined by the use of soil amendants with plant growth and development. There is a lack of information on the growth parameters usually sought in such types of studies, which are absent in this work. In addition, it is normal that if the number of plants per hectare is higher, the overall yield is potentially higher even if the yield per plant does not increase.  There are two possible effects at play here: those on germination and that on yield per plant, so it would have been important to carry out germination tests on the extracts of organic ammentants and to report productivity data per plant.

In conclusion, I consider the interpretation of the results obtained to be somewhat limited, confusing, and repetitive, and therefore needs to be modified and implemented in light of the above. I also suggest that it might be better to include the discussion, interpretation and comparison with adequate bibliographic citations directly after the presentation of the results, and then to report an overall view of the study carried out in the Conclusions, in which the new and improved aspects brought about by this study on highly topical arguments should be highlighted more clearly.

Author Response

(The authors gave the same response as above.)

Round 2

Reviewer 1 Report

Comments and Suggestions for Authors I agree with the changes made to the revised manuscript. I think that the manuscript has become much better. However, the authors gave a very formal answer to one of my remarks: "For Tables 3-6, it would be useful to provide LSD values. In the tables presented, it is not always obvious why different letters are used for averages with a small difference, while the same letters are used for averages with a large difference. For example, TOC and P2O5 are in Table 4, and Phosphatase is in Table 6.” . Answer: We did not provide more details to keep the table clear of additional data that could clutter it. O.K. But I have doubts about the correctness of using letters as a result of statistical analysis for the indicators “TOC” and “P2O5” in Table 4. What were the LSD values for these parameters? I need this information as a reviewer to verify the table values.

Author Response

Dear Reviewer, thank you for your valuable comments. The LSD values were applied to the corresponding tables.

Reviewer 2 Report

Comments and Suggestions for Authors

I greatly appreciated the revision work carried out by the authors, but I must once again point out the presence of several shortcomings that need to be corrected, particularly with regard to the Materials and Methods section. I note a certain superficiality in the description of the procedures, with respect to the analysis of the chemical and chemico-physical parameters of the various matrices investigated.  Below are the points that, in my opinion, are still critical and need to be improved:

_ in line 117 you have to correct "blick" with "block"

_ in line 133 authors must add the percentage composition of cow manure and wheat straw in compost

_ the authors' reply to my comment 6 stating "The aim was to compare the usual amounts of mineral fertilisers used in southern Bulgaria for growing barley on alluvial-meadow soils with the organic amendments introduced in quantities in the low concentration spectrum of those applied in the literature" needs further clarification and in any case this important aspect must be highlighted as part of the text of the paper. I believe it would be extremely useful, for a better understanding of the experimental design, to add a new table in which the quantities of the main macronutrients added in total in the five different treatments are compared.

_ section 2.2 still lacks some information on the analyses performed on vermicompost and biochar, for which the authors only refer to the literature: it is important to always include a brief description of the analytical procedures, providing information about the chemicals and instrumentation used, and then cite appropriate articles for further study; in this way, among other things, it also becomes clear which parameter is actually being quantified where there may perhaps be ambiguous interpretations; for example, in the article mentioned for the analysis of vermicompost (ref.28), several methods are described in comparison to quantify organic carbon, and depending on the method, a different meaning is given to the parameter. For the same reason, it is not clear whether P and K in Table 1 means total or available P and K. Reference 29 cited for the analysis procedures on Biochar, is not appropriate as it only describes the analysis methods for total N (N-Tot) and total C (TC) by means of an elemental analyser among the parameters in Table 1. Is this so? Which instrument was used? And for the other parameters?

_ Section 2.3.1 Soil analyses still needs some further clarification:

i) as it is described, the procedure for the determination of accessible soil nitrogen seems to be limited to the analysis of N-NH4 alone, as only distillation and final titration of the extract is mentioned. Nothing is said about the quantification of the other soluble forms of nitrogen (NO3 and NO2). If this were indeed the case, then the figure given in Tables 2 and 4 should be indicated as N-NH4 and not N, which would, on the contrary, imply the total extractable nitrogen, which is the sum of the three different forms and which is by far the most useful data.

ii) the procedure for the analysis of available phosphorus also continues to be lacking: apart from the lack of a brief description, two references (34,35) are cited in which the Egner-Reihm method used by the authors should be illustrated. Both of the works mentioned are not appropriate: the first is dated and cannot be found on the web, while the second manual mentioned contains several methods for analysing available phosphorus (Bray and Kurtz method, Olsen method, etc.) but not the one used by the authors. 

_ Section 2.3.1 Plant tissue analyses also needs clarification:

i) relevant bibliographical references to the methods used for all parameters (total N, total P and total K) on this type of matrix are missing;

ii) the necessary information regarding the quality of the chemicals used (brand, purity, etc.) is not reported, but this deficiency is uniquivocal throughout the Materials and Methods section;

iii) hydrogen peroxide is erroneously reported as the catalyst when in fact it is a strong oxidant: the official Kjeldhal method uses a copper or selenium salt as the catalyst, and requires the presence of potassium sulphate to raise the boiling point of the sulphuric acid;

iv) the method chosen to quantify, after distillation, the ammonia released is omitted;

_ In Line 228, after "...the method described by...." the author's name must be added. 

_ In Table 3, the authors made the requested correction: but how come from P2O5 to P and from K2O to K the numerical values did not change?

_ Pay attention to the significant digit of values entered (mean and standard deviation) in table2.

_ In lines 243-244 the authors state that "The amendments' application affected the soil's physicochemical properties (Table 5). The soil pH in the rhizosphere was reduced from 8.2 to less than 8". How can this sentence be reconciled with the fact that in reality the greatest decrease in pH between the soil harvested before the experiment and at the end of the experiment concerns the control soil (pH8.2 versus pH7.58 respectively) where no soil conditioner or fertiliser was added? How is this interpreted? The authors should disquisition this in Discussion at Lines 389-392, where the problem is approached

Author Response

Dear Reviewer, thank you for your additional observations and comments. Below, you can see their answers.

Kind regards,

Stefan Shilev
